# The *Asanté™* HIV-1 *Rapid Recency®* Assay is reliable, feasible, and acceptable for use at the point-of-care in Lusaka, Zambia

Shilpa S. Iyer[1], Jake M. Pry[1,2]*, Herbert Kapesa[1], Misinzo Moono[1], Chilambwe Mwila[1], Christiana Frimpong[1], Mirriam Nanyangwe[1], Lumbani Phiri[1], Ruth Ngandu[1], Precious Sakanya[1], Sharon Mwansa[1], Talandila Phiri[1], Mizinga Haciwa[1], Patricia Maritim[1], Kemba Lee[3], Melissa Arons[3], Tiffiany Aholou[3], Peter Minchella[4], Theodora Savory - van Huis[1], Carolyn Bolton[1,5], Michael E. Herce[1,6]

1 Centre for Infectious Disease Research in Zambia, Lusaka, Zambia, 2 University of California, Davis, California, United States of America, 3 Division of Global HIV & TB, Global Health Center, Centers for Disease Control and Prevention, Atlanta, Georgia, United States of America, 4 Division of Global HIV & TB, Global Health Center, Centers for Disease Control and Prevention, Namibia, Windhoek, Namibia, 5 University of Alabama, Birmingham, Alabama, United States of America, 6 University of North Carolina, Chapel Hill, North Carolina, United States of America

* jmpry@ucdavis.edu

## Abstract

### Background

Zambia established a recent infection testing algorithm (RITA) incorporating a novel point-of-care (POC) rapid test—the Asanté™ HIV-1 Rapid Recency® Assay (RTRI)—plus a HIV-1 viral load (VL) test to distinguish recent (≤12 months) from long-term (>12 months) HIV acquisition. This study evaluated the field performance of RTRI when implemented by healthcare workers at the POC.

### Methods

We enrolled individuals newly diagnosed with HIV between 20 May 2021 and 10 March 2022 at two Ministry of Health facilities in Lusaka, Zambia. Participants received on-site RTRI testing and provided an additional sample for repeat RTRI and VL testing at a central laboratory. Final recent infection testing algorithm (RITA) results were returned to the study sites and were made available to clients at their study follow-up visit. Agreement between POC- and laboratory-RTRI was assessed using Cohen's Kappa. We compared recent versus long-term HIV classification across testing locations using the national RITA as the reference standard. Four focus group discussions (FGDs) with health staff explored perceptions surrounding POC-RTRI implementation.

**Data availability statement:** De-identified and aggregated study data are publicly available within the Dryad data repository. DOI: 10.5061/dryad.k98sf7mnf.

**Funding:** The recency program was financially supported by the President's Emergency Plan for AIDS Relief (PEPFAR) through the Centers for Disease Control and Prevention (CDC) under the terms of a cooperative agreement with CIDRZ (NU2GGH001920). The findings and conclusions in this report are those of the author(s) and do not necessarily represent the official position of CDC.

**Competing interests:** None to declare.

## Results

Agreement between POC and laboratory RTRI was 96.5%, with a Kappa of 0.812 (95% CI: 0.704–0.920). The POC-RTRI results indicated numerically more recent infections than laboratory-RTRI (30 vs 27), with three POC-RTRI false positives resulting in reduced sensitivity 85.0% for the POC-RTRI compared to 100.0% sensitivity for the laboratory-RTRI against the RITA reference standard. FGD participants (n = 28) agreed that POC RTRI was feasible and acceptable with adequate training, human resources, client counselling, and quality assurance measures.

## Conclusion

There was strong concordance between POC- and laboratory-RTRI results. The findings support the feasibility of implementing RTRI at POC by non-laboratory health workers, provided adequate training and health system resources are in place.

## Introduction

Despite increasing availability of HIV testing service (HTS) globally, important gaps remain in ensuring all people living with HIV (PLHIV) know their HIV status, particularly in high prevalence countries in sub-Saharan Africa (SSA). In Zambia, results from the most recent population-based HIV/AIDS impact assessment (ZamPHIA), completed in 2021, revealed that approximately 11% of PLHIV are unaware of their HIV status [1]. These data suggest a need for scale up of varied, innovative, and evidence-based HTS platforms such as index testing. While index testing has been a successful strategy for newly diagnosing PLHIV, the current practice of engaging sexual and needle-sharing contacts of persons living with, or at high risk of, HIV without regard to HIV viremia or the phase of infection (i.e., chronic versus acute and early HIV infection [AEHI]), may be insufficient, in and of itself, to increase program gains [2].

Indeed, AEHI has complicated efforts to achieve HIV epidemic control through high replication of viruses with increased infectivity [3,4], disproportionately contributing to onward HIV transmission, particularly during the first 6 months after HIV acquisition [5]. Yet, AEHI is hard to identify in real-world program settings where availability of viral load (VL) or combination antigen/ antibody testing at the point-of-care (POC) is limited for this purpose [6]. In the absence of these POC technologies and other tools that can routinely identify AEHI for public health intervention, interest has grown in developing new assays that can reliably differentiate between individuals with recent HIV infection (i.e., occurring within 6–12 months of HIV acquisition, including AEHI) and those with more chronic infection (i.e., occurring after 12 months from acquisition). Such tests, called tests for recent infection, cannot be used to diagnose AEHI, but may still have utility in national HIV surveillance programs intended to identify geographic areas of potentially ongoing, and increased, HIV transmission to inform a precise public health response [1]. One such test, the *Asanté* HIV-1 Rapid Recency test (Sedia Biosciences, Oregon, USA), is a POC rapid test for recent infection

(RTRI) developed by the U.S. Centers for Disease Control and Prevention (CDC) to differentiate, under research conditions, people with long-term HIV infection from those with seroconversion in the last 6–12 months [7,8]. Results from these tests are often used in combination with one or more of VL, CD4 + count, ART history, plasma antiretroviral metabolites, and/or clinical evaluation for AIDS-defining illnesses as part of a recent infection testing algorithm (RITA) to assess HIV recent infection status.

In Zambia, the Asanté RTRI is utilized as part of national surveillance, and test results are not integrated into patient-level care. Under the Zambian surveillance protocol, newly diagnosed PLHIV receive an RTRI and VL test conducted at a central laboratory and deidentified results are used in aggregate to identify geographies with a potential for increased or ongoing HIV transmission (i.e., "hotspots"). In principle, at a programmatic level, identification of such geographies allows for focusing prevention resources on high-transmission geographical areas to potentially interrupt active transmission. However, the utility of the RTRI for this purpose remains an area of ongoing debate as it is unclear whether it accurately classifies individuals as having recent HIV acquisition when used in isolation, thereby limiting the potential of the test [9–11]. Multiple studies have assessed the accuracy of RTRI classification, compared to other laboratory-based assays that detect low-avidity antibodies and robust algorithms that incorporate clinical and laboratory data, but these assays have not yet received U.S. Food and Drug Administration (FDA) approval [12,13]. In theory, adoption of RTRI at the POC could enable faster time to results and public health action at facility-level, while saving on resources for sample shipping and specialized laboratory personnel. However, no published studies from sub-Saharan Africa (SSA) have examined the effects of the Asanté RTRI when used at the POC by non-laboratory healthcare workers on testing accuracy, reliability, or site-level operations.

In this study, we sought to assess the operational feasibility and reliability of the Asanté RTRI when used by non-laboratory healthcare workers at the POC, and to evaluate the frequency of discrepant results by RTRI testing setting (i.e., POC versus centralized laboratory). Using additional laboratory results, including VL testing, we also evaluated the field performance of the Asanté RTRI when used in the central laboratory and at the POC compared to the program reference standard used in Zambia, the central laboratory-based RITA. Lastly, through focus group discussions (FGDs) with front-line healthcare workers (HCW), we sought to assess the feasibility of implementing Asanté RTRI at the POC in Zambian health facilities.

## Methods

### Study design

This is the laboratory evaluation component of an implementation research study called REACHZ (*R*ecency *E*nhanced *A*lgorithm for *C*ontrolling *H*IV in *Z*ambia) designed to describe the effects of integrating RTRI testing at the POC on the performance of routine index testing within Zambia's PEPFAR-supported national HIV program. The laboratory evaluation presented here incorporates both quantitative and qualitative components using a convergent, parallel mixed methods approach. For the quantitative component, we assessed field performance of the Asanté RTRI when deployed at the POC compared to testing done at a central laboratory and compared both to the RITA as the reference standard. For the qualitative component, we conducted FGDs with HCWs and other stakeholders to evaluate the feasibility of POC-RTRI testing according to empirically supported frameworks [14–16].

### Study setting and population

The REACHZ study was conducted at two high-volume first-level Ministry of Health (MOH) hospitals supported by CDC/PEPFAR (i.e., sites A and B) in the heart of urban Lusaka, Zambia between 20 May 2021—10 March 2022. The study population included adults (≥18 years of age) presenting as new to HIV care who consented to having an additional blood sample taken for recency and viral load testing.

## Study procedures

Potential participants included all individuals accessing routine HTS and newly diagnosed with HIV at a study clinic. These potential participants were invited by study staff to take part in the study. Two study peer-educators sensitized potential participants about the study and referred interested individuals to trained research assistants for informed consent procedures. After obtaining written informed consent, study enrolment procedures were conducted, including completion of a locator form, enrolment demographic case reporting form, and biobehavioral survey. Study staff conducted all study-specific procedures, including accompanying participants to clinic rooms for recency testing and sample collection. Once in the clinic room, HCWs trained by the study performed a finger prick on consenting participants for POC-RTRI testing. The PI and study laboratory specialist trained four facility-based non-laboratory HCWs on how to perform and interpret the Asanté RTRI at the POC and how to use quality control specimens on new test kits to assure their performance. The PI and laboratory specialist also employed proficiency panels to assess tester competency using materials from the national surveillance program to mirror routine testing conditions. Finally, the study trained one MOH phlebotomist at each site to collect one venous blood sample from each participant for confirmatory RTRI and VL testing per the national RITA done at the Centre for Infectious Disease Research in Zambia (CIDRZ) Central Laboratory in Lusaka (henceforth referred to as the "central laboratory") (see *Test procedures* below). Results from all POC and central laboratory testing were returned to study staff and participants at scheduled study follow-up visits.

## Laboratory and test procedures

The Asanté is a lateral flow assay with three lines per test strip, including control (C), HIV-positive serostatus verification (V), and long-term (LT) lines that help distinguish recent from long-term infection [7], and can be read qualitatively by visual strip inspection. Trained facility-based non-laboratory HCWs performed the Asanté RTRI at the POC at the two study sites. To allow RTRI result confirmation and VL testing, an additional 5 ml blood sample was collected from each participant. In this way, blood samples were tested twice on the Asanté RTRI, once at the facility and a second time at the central laboratory. Testers recorded all observed lines by drawing the observed lines on a picture replica of the test on a structured study form. This allowed documentation of test strip line interpretation, since the absence of the LT line indicates a recent result and may lead to incorrect classification. The lines drawn on the picture were verified to ensure they corresponded to the result documented through regular quality control (QC) record reviews done by the research assistants and study laboratory specialist during supportive supervision visits. Results from the POC RTRI were recorded at the site and not released to the participant until confirmation from the central laboratory was obtained. In cases of discrepant results, the central laboratory classification was used as the gold standard. Proficiency testing of the testers to assess tester competency was performed as part of study start up training. Given the short time frame for study implementation and interruptions due to COVID-19, study testers did not participate in the routine semi-annual proficiency testing under the recency surveillance program after the initial proficiency testing.

A combination of in-person site visits, study record reviews, and tele-monitoring were used to regularly assess testing quality throughout the study period. The study laboratory specialist conducted in-person supportive supervision visits at each study site at least twice weekly. The aim of supportive supervision visits was to review site records for QC, resolve any outstanding QC queries, provide logistical support, and collect process data to assess the quality of study implementation and adherence to study standard operating procedures. QC reviews were guided by a study QC checklist and reported on a QC log, both of which were contained in each participant file. The QC checklist listed all the study procedures and records expected according to the study protocol. The QC log captured information about the QC query including the dates of review and resolution, description of the query and its resolution and initials of the reviewer and resolver.

Telemonitoring was done daily for the purposes of taking corrective action and clarifying faint band lines on the RTRI tests. Telemonitoring for RTRI test interpretation and adjudication involved the RTRI tester photographing test strips using a standard protocol (i.e., within the specified test reading time – 20 minutes, using a white background and natural light).

Photographs were uploaded to a secure, encrypted study virtual platform restricted to study personnel. The study laboratory specialist and principal investigator (PI), blinded to participant identifiers, independently reviewed the images within 24 hours. A band was confirmed present if both reviewers confirmed visibility of noticeable bands and absent if either reviewer deemed it unnoticeable. In cases of disagreement, a third reviewer – a trained laboratory technologist from the CIDRZ central laboratory – provided final interpretation. Any observed band, even if faint, was recorded as present and the sample was considered as long-term for the purposes of this study.

The VLs were quantified for all samples with a recent RTRI result (as determined by the central laboratory RTRI) and a subset of samples with a long-term RTRI using the Aptima® HIV-1 Quant Assay on the Hologic Panther platform (Marlborough, Massachusetts, USA). A trained central laboratory technician performed VL testing and conducted internal QC procedures according to established central laboratory standard operating procedures. VL results were used in combination with Asanté RTRI results to classify the samples according to national RITA definitions (below). A recent result on the RTRI accompanied by a suppressed viral load was considered as a false-recent result and counted as long-term.

## Case definitions for recent and long-term HIV infection

We classified the duration of HIV infection according to the national RITA in Zambia, and classified an individual as having RITA-recent infection if they did not disclose prior HIV diagnosis or ART exposure, had evidence of recent HIV acquisition on the Asanté RTRI, and had a VL ≥ 1,000 copies/ml. A RITA long-term HIV infection was defined based on a long-term result on the Asanté RTRI, irrespective of VL. RITA long-term status was also assigned to a person living with HIV if they had evidence of recent HIV acquisition on the Asanté RTRI but were subsequently found to have a VL < 1,000 copies/ml. VL results are included in the case definition to improve the accuracy of recent infection classification per the national RITA.

## Qualitative data collection and analysis

We conducted focus group discussions (FGDs) with frontline HCWs involved in voluntary HIV counselling and testing, provider-initiated testing and counseling, index testing, recency testing, and/or ART initiation at the two study sites to describe implementation and health system issues that might affect the feasibility of Asanté RTRI testing at the POC. HCW cadres were classified as peer treatment supporters (also known as community health workers), nurses, clinical officers, HIV testing service counselors, and laboratory technicians. FGD participants included a consecutive sample of 4–9 eligible MOH HCWs who responded to an open invitation to join the FGD at each site. Eligible FGD participants were ≥18 years of age, a professional or community HCW, and involved with HTS or other form of HIV service delivery at a study site. Following informed consent, two trained female FGD qualitative researchers conducted audio-recorded FGDs using a semi-structured discussion guide. The FGD guide explored provider implementation experiences in line with the study objectives and informed by Proctor's Implementation Outcomes Framework [14]. We transcribed and translated FGD audio-recordings into English in one step incorporating expanded field notes.

Guided by Proctor's Framework and using thematic analysis, we focused largely on feasibility and acceptability. These steps included familiarization, coding, generating initial themes, reviewing and developing themes, refining, defining and naming themes and reporting. Feasibility was defined as the extent to which POC RTRI testing could be successfully used within routine health facility settings [14–16]. We also focused on assessing provider's reactions to RTRI testing (acceptability), provider's use of the test and the extent to which providers thought the RTRI testing could be delivered in light of available resources (appropriateness), and provider's capacity to deliver the tests as intended (implementation) [17]. Data analysis begun by familiarization with the FGD transcripts. Coding was conducted through a combination of deductive approaches based on the *a priori* themes described above, as well as inductively to capture additional themes highlighted by the providers. Data management was done in NVivo QSR™ (Queensland, Australia). After two transcripts had been coded, the research team met to hold team discussions and review key emerging themes to ensure there was consensus

with the coding framework before applying the codes broadly across all the transcripts [18]. Key findings relating to the research question are presented below with supporting evidence in extracted quotes.

### Sample size considerations and quantitative data analysis

For this descriptive implementation research study that did not involve *a priori* hypothesis testing, we did not specify a sample size. Rather, we estimated a feasible enrollment target based on routine HIV testing data at our proposed study sites for the duration of the planned study. As such, we estimated that we could approach a total of 50 clients newly identified with HIV per week across both study sites, of whom approximately 50% would meet study eligibility criteria, leaving an estimated recruitment target of 600 over the 24-week planned study period. Due to unforeseen challenges emerging because of the COVID-19 pandemic, routine clinic visit volumes reduced by roughly 50% across Lusaka during the study period, leaving us with a project study enrollment of about 300 participants.

We used descriptive statistics to summarize study participant characteristics. We then assessed *Asanté™* HIV-1 *Rapid Recency® Assay* performance at the POC compared to the central laboratory by comparing the classifications of recent and long-term infections in each setting using as our reference standard the recency classification from the central laboratory-based RITA, which is the program reference standard used in Zambia. We also calculated the Cohen's Kappa statistic to estimate agreement between the *Asanté™* HIV-1 *Rapid Recency® Assay* performed by non-laboratory HCWs at the POC and technicians at the central laboratory. All analyses were conducted in Stata v17 (StataCorp, College Station, Texas, USA).

### Ethics

The study protocol was approved by the University of Zambia Biomedical Research Ethics Committee (#1157–2020), the IRB of the University of Alabama, Birmingham, USA (#IRB-300006066) and performed in line with the principles of the Declaration of Helsinki. This activity was reviewed by CDC, deemed research not involving human subjects, and was conducted consistent with applicable federal law and CDC policy.

## Results

### Overview

For the quantitative component, we invited a total of 393 of 1,224 (32.1%) people newly diagnosed with HIV at the two study sites during the study period for study participation. Of these, 344 (87.5%) were found study eligible and 321 (93.3%) enrolled. Of the 23 (6.7%) who were eligible but did not enroll, 14 (60.9%) stated that they did not have enough time to participate, 6 (26.1%) refused informed consent, 2 (8.7%) were not interested in study participation, and 1 (4.3%) did not complete study enrollment procedures. For this analysis, we included 317 of 321 (98.8%) participants after excluding 4 (1.2%) participants—all of whom had a negative HIV-positive serostatus verification (V) line at both POC and central laboratory (or in one case, just at the central laboratory). We assessed Asanté RTRI test performance characteristics in a subset of 245 (76.3%) participants with complete VL data (Fig 1).

### Performance of the RTRI HIV diagnostic verification line (positive verification line) among confirmed HIV-seropositive specimens by the national HIV diagnostic testing algorithm

A total of 321 plasma specimens from persons diagnosed with HIV infection using the serial national diagnostic testing algorithm were also tested by the Asanté RTRI. All 321 samples (100%) were tested at both the facility and the central laboratory, having been received at the central laboratory with complete sample information. For 321 (100%) of these tests, visible control lines were observed. In 3 (0.9%) instances, the verification line indicating HIV antibody seropositivity was not observed on Asanté RTRIs done at both the POC and central laboratory. These samples also did not have a

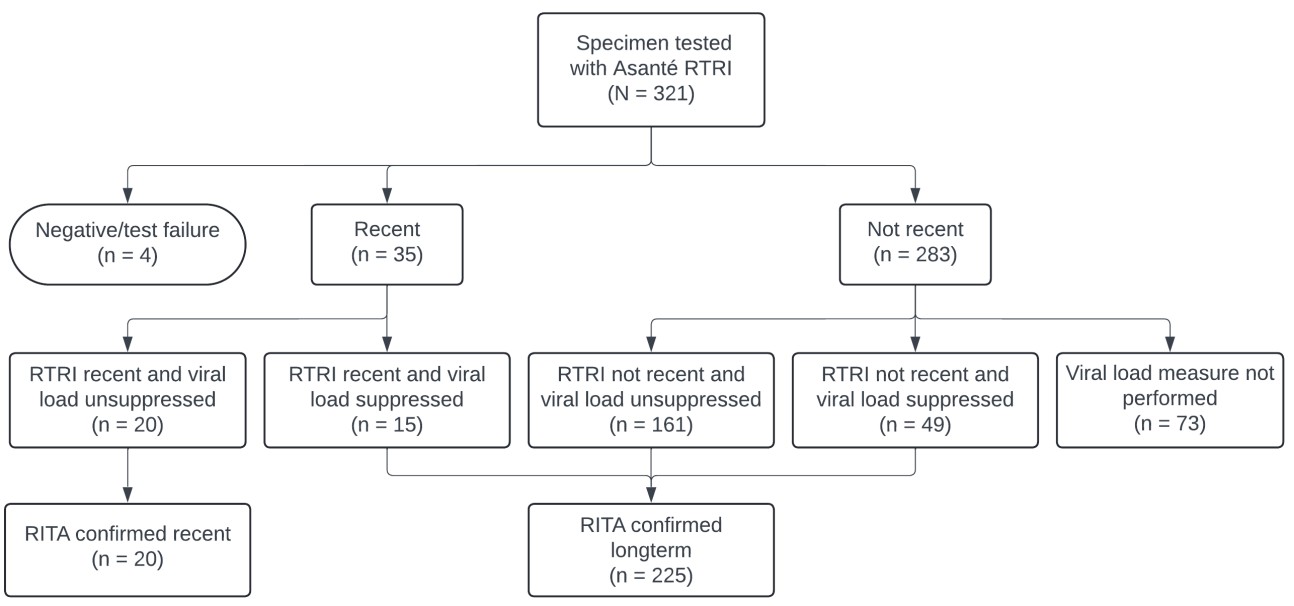

**Fig 1. Summary of participant recency testing\*.** \*Based on Asanté RTRI test results from the central laboratory (CL).

quantifiable plasma VL, ruling out acute HIV-1 infection. Thus, we concluded that these 3 (i.e., 3 of 321 samples) samples had discrepant results between the national serial HIV diagnostic testing algorithm and the Asanté RTRI tests performed through the study. In one additional case, an HIV-positive verification line was present at POC but did not develop on the test strip used at the central laboratory despite a measurable VL for the specimen, likely indicating a non-reactive central laboratory Asanté RTRI test strip. Thus, 4 of 321 cases were excluded from the analysis.

### Agreement between Asanté RTRI tests done at the Central laboratory and POC

A total of 317 (98.8%) samples with evident control and HIV-positive verification lines were used to assess performance of the Asanté RTRI done by a HCW at the POC compared to a test done in parallel by a laboratory worker at the central laboratory. Of 30 specimens classified as recent by HCWs, 27 (90.0%) were similarly classified as recent by the central laboratory. Of the 287 specimens classified as long-term at the POC, 279 (97.2%) were similarly classified by the central laboratory. Overall, 306 of 317 (96.5%) specimens had concordant results for RTRI tests done at the central laboratory and POC. This comparison yielded 96.5% agreement and a Kappa statistic of 0.812 (0.704–0.920), indicating substantial agreement between Asanté RTRI tests performed by trained HCWs at POC and laboratory staff at the central laboratory. Of the concordant results, 279 were long-term concordant results and 27 were recent concordant results (Table 1). To assess for tester-specific differences in concordance, we analyzed the data by enrollment site. The concordance of RTRI results was 94.1% at Facility A and 95.5% at Facility B.

We identified 11 discordant results. Of these, eight and three were classified as recent and long-term by the central laboratory, respectively, disagreeing with the POC classification. Using the lines drawn on result reporting forms, we confirmed that result discordance was not due to incorrect test performance, but, instead, due to inaccurate interpretation of visible bands on the RTRI done at the POC. These discordant results were observed at the start of the study. By strengthening a combination of quality control measures, including proficiency testing for HCWs and telehealth and in-person site monitoring visits, test discrepancies were subsequently resolved.

**Table 1. Concordance of Asanté RTRI testing at the POC and Central Laboratory (N = 317).**

| Asanté RTRI at the CL | Asanté RTRI at the POC | | |
|---|---|---|---|
| | Recent | Long-term | Total |
| Recent | 27 | 8 | 35 |
| Long-term | 3 | 279 | 282 |
| Total | 30 | 287 | 317 |
| Agreement (%) | 96.5% | | |
| Kappa | 0.812 (95% CI: 0.704–0.920) | | |

Note: negative results excluded; RTRI = rapid test for recent infection, CL = central laboratory, POC = point-of-care.

### Asanté RTRI test results at POC compared to RITA results done at the central laboratory

A total of 245 (76.3%, 245/321) participants with an HIV-positive result and complete RTRI and complete VL data contributed to our accuracy analysis (Table 2). Participants had a median age of 30 years (interquartile range, IQR: 25−37 years), 63.3% were female, and 33.1% enrolled at Clinic A, while 66.9% enrolled at Clinic B. Due to logistical reasons and the COVID-19 delta variant wave in July 2021, study participants were not enrolled in that month.

It is known that tests for recent infection can misclassify individuals with long-term infection as having recent infection, particularly individuals on suppressive ART or elite controllers with low VL, which can cause a loss or delay in avidity or maturation of antibodies used as the basis for differentiation by assays like the Asanté RTRI.

Using data from the subset of participants with a VL, we assessed the classification of HIV infection into recency categories according to the national RITA, namely, "long-term," "confirmed long-term," and "confirmed recent" (Fig 1). A total of 35 participants (10.9%, 35/321) had their infection initially classified as recent, based on Asanté RTRI results from the central laboratory. Of participants with a recent RTRI test result, 20 (57.1%, 20/35) had unsuppressed VL (i.e., ≥ 1000 copies/ml), and were considered to have confirmed recent infection by the national RITA. As seen in Fig 1, out of 35 PLHIV with a recent RTRI result, 15 (42.9%, 15/35) had a suppressed VL (i.e., < 1000 copies/ml) and had their infection status re-classified by the RITA.

Using the RITA result as a reference standard, we observed superior performance of the Asanté in the central laboratory setting per receiver operator curve (ROC), with 0.9667 area under the curve (95% confidence interval [CI]: 0.9457, 0.9877) as compared to the POC setting, which had a smaller 0.8983 area under the curve (95% CI: 0.8054, 0.9913) (Fig 2 and Tables 3, S2 and S3 in S1 File). Correct classification, defined as those observations where the point of care RTRI or the central laboratory RTRI test and the RITA result agree, was 94.0% for both central laboratory and POC RTRI results (Fig 2, Tables 3, S2 and S3 in S1 File). In a sensitivity analysis in which we excluded observations where baseline viral load was suppressed (≤1000 copies/mL), correct classification increases to 100.0% for central laboratory RTRI and 97.8% for POC RTRI (Figure S1 and Table S1 in S1 File).

Median VL among PLHIV with confirmed recent infection was 37,335.5 copies/mL (IQR: 13,558–313,757 copies/mL) and 43,680 copies/mL (IQR: 305−266,789 copies/mL) among PLHIV with confirmed long-term infection. Of PLHIV identified as having long-term infection, 71.6% (95% CI: 65.2, 77.4%) had an unsuppressed viral load (≥1,000 copies/mL).

### Qualitative results

For the qualitative component, we enrolled 28 participants into 4 focus group discussions (FGDs). Each FGD was made up of 5–9 HCW participants drawn from various cadres (Table 4). We analyzed the FGD transcripts and described the extent to which the providers considered RTRI implementation at the POC to be feasible, acceptable, and appropriate in the following section.

**Table 2. Participant characteristics by recent infection testing algorithm (RITA) status (N = 245).**

| Factor | Level | Total | Long-term | Confirmed recent | p-value |
|---|---|---|---|---|---|
| | | n (%) | n (%) | n (%) | |
| N | | 245 | 225 | 20 | |
| Sex of the participant | Female | 155 (63.3) | 141 (62.7) | 14 (70.0) | 0.514** |
| | Male | 90 (36.7) | 84 (37.3) | 6 (30.0) | |
| Age | median (IQR) | 30 (25-37) | 30 (25-36) | 28.5 (24.5-37) | 0.574† |
| Age category | 18-24 years | 55 (22.4) | 50 (22.2) | 5 (25.0) | 0.948* |
| | 25-34 years | 107 (43.7) | 99 (44.0) | 8 (40.0) | |
| | 35-44 years | 68 (27.8) | 62 (27.6) | 6 (30.0) | |
| | 45 + years | 15 (6.1) | 14 (6.2) | 1 (5.0) | |
| Marital status | Single | 38 (15.5) | 34 (15.1) | 4 (20.0) | 0.582* |
| | Married | 115 (46.9) | 108 (48.0) | 7 (35.0) | |
| | Divorced/Separated | 83 (33.9) | 75 (33.3) | 8 (40.0) | |
| | Widowed | 9 (3.7) | 8 (3.6) | 1 (5.0) | |
| Educational attainment | No formal education | 4 (1.6) | 4 (1.8) | 0 (0.0) | 0.765* |
| | Primary | 58 (23.7) | 54 (24.0) | 4 (20.0) | |
| | Secondary | 168 (68.6) | 154 (68.4) | 14 (70.0) | |
| | College/University | 15 (6.1) | 13 (5.8) | 2 (10.0) | |
| Virally suppressed (<1000 copies/mL) | Unsuppressed | 181 (73.9) | 161 (71.6) | 20 (100.0) | 0.003* |
| | Suppressed | 64 (26.1) | 64 (28.4) | 0 (0.0) | |
| Healthcare facility | Clinic A | 81 (33.1) | 70 (31.1) | 11 (55.0) | 0.030** |
| | Clinic B | 164 (66.9) | 155 (68.9) | 9 (45.0) | |
| Month of study enrollment | June 2021 | 8 (3.3) | 6 (2.7) | 2 (10.0) | 0.182* |
| | July 2021 | 0 (0) | 0 (0) | 0 (0) | |
| | August 2021 | 11 (4.5) | 8 (3.6) | 3 (15.0) | |
| | September 2021 | 65 (26.5) | 59 (26.2) | 6 (30.0) | |
| | October 2021 | 35 (14.3) | 32 (14.2) | 3 (15.0) | |
| | November 2021 | 21 (8.6) | 20 (8.9) | 1 (5.0) | |
| | December 2021 | 12 (4.9) | 12 (5.3) | 0 (0.0) | |
| | January 2022 | 47 (19.2) | 44 (19.6) | 3 (15.0) | |
| | February 2022 | 46 (18.8) | 44 (19.6) | 2 (10.0) | |

Note: IQR – interquartile range, † - T-Test, * - Fisher's Exact Test, ** - Chi-squared Test.

## Feasibility and acceptability

Facility healthcare providers and peers, responsible for the implementation of the RTRI, found it feasible and acceptable to implement at the POC. In relation to the perceived efforts required to implement the test, and the actual fit with their routine practices, providers felt that the test itself was not difficult to implement or disruptive to their workflow. One clinical officer mentioned,

> *"It has not made our work day harder I would say because when we are doing the sample collection for all the clients for the recency, we will do baseline sample collections. So, it's just an addition if we are collecting for recency. It is just additional to the routine that we are* [already] *doing".* (Site A Clinician)

In some cases, RTRI testing was perceived as helping address questions arising from clients during HCW's daily work, as one health facility peer treatment supporter noted,

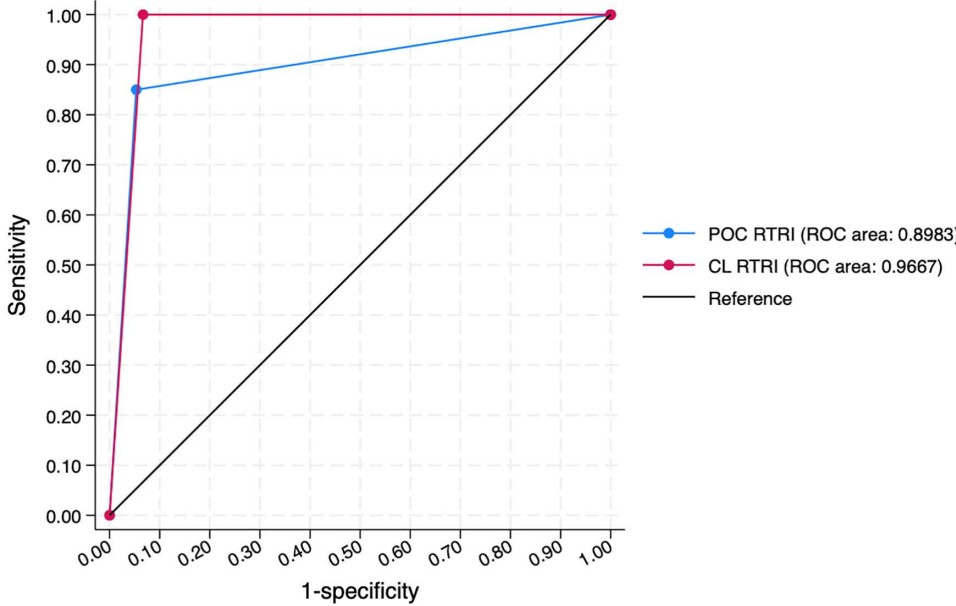

**Fig 2. Receiver-operator curve and test characteristics for the Asanté when conducted at the point-of-care and in a central laboratory, compared to the RITA used in Zambia as a reference standard (N = 245).** Note: RTRI – rapid test for recent infection, POC – point of care, ROC – receiver-operator curve, CL – central laboratory.

**Table 3. Rapid test for recent infection classification characteristics by test setting compared to the Zambia national program recent infection testing algorithm as the reference standard.**

| Test setting | Sensitivity (compared to RITA) | Specificity (compared to RITA) | Correctly classified |
|---|---|---|---|
| CL | 100.0% | 93.4% | 94.0% |
| POC | 85.0% | 94.7% | 94.0% |

Note: CL – central laboratory, POC – point-of-care, RITA- recent infection testing algorithm.

**Table 4. Characteristics of healthcare worker focus group participants (N = 28).**

| Characteristics | | Count n (%) |
|---|---|---|
| Sex | Female | 15 (53.6) |
| | Male | 13 (46.4) |
| Site | A | 14 (50.0) |
| | B | 14 (50.0) |
| Healthcare worker cadre | Peer treatment supporter/ community health worker | 17 (60.7) |
| | Nurse | 6 (21.4) |
| | Clinical Officer | 2 (7.1) |
| | HTS Counselor | 2 (7.1) |
| | Laboratory Technician | 1 (3.6) |

*"Another problem that used to be there…was when a person asks you where they got the virus from. Because* [they would say]*, '*[the] *last time when I tested I was found to be negative and this time I have been found positive'. They ask those questions and now you are trying by all means to help your client so that they can come to terms with their ehhh status. So recency has been found to help* [us] *give an answer to clients that have such questions."* (Site B, Peers).

Initial resistance to POC implementation of the RTRI among providers was driven by a lack of understanding of how the service would fit into their regular activities. Training changed their knowledge and beliefs about the RTRI, which improved their satisfaction with the practice. Thus, healthcare workers highlighted the importance of training and supervision to ensure the sustained adoption and accurate use of the RTRI at POC.

*"Just around the facility, we have capable people who can teach others on good skills* [for recency testing]*. So, we have deliberately said we will be doing in-house training where we don't have to call an outsider to come and… take them through the skills. And so that* [way] *we have done* [RTRI testing] *within ourselves and then it is helping."* (Site B Healthcare Provider)

### Appropriateness

Due to the increased number of samples required from the participating clients, the providers highlighted the importance of sensitizing clients on the additional samples, time and processes needed for the various required tests at the time of establishing HIV care. This sensitization was needed to improve the suitability of POC RTRI testing for routine clinical practice.

*"The challenge was the bleeding of the client because of that additional one sample that we were collecting, they were complaining to say why we were collecting so many bottles, in as much as we might try to explain that there is a recency program… on that* [new] *process, clients, just to convince them otherwise it was a challenge."* (Site B, Health care worker)

Providers reported a good level of compatibility between the test and other HIV testing services being offered at their health facilities, which enabled its successful implementation.

*"So, because* [HIV index testing] *clients are always there, they* [health care workers] *want to be able to turn around* [recency test results] *when we do it in the quickest time possible..."* (Site B, Healthcare provider)

*"When starting* [recency testing]*, I thought that they just wanted to disturb our brains, but after we knew how recency works we realized that it helps us a lot and that's why the* [number of] *positives are increasing."* (Site B, Peers).

Despite the RTRI being perceived as being aligned with existing facility structures, systems, and procedures, potential implementation of POC testing would need to consider additional supports and resources. In particular, there was an identified need for more human resources due to the additional steps occurring outside of routine HIV testing, as well as training in areas such as contact elicitation for index testing.

*"Our workload has increased because it means more registers coming in, it means more clients coming in, and clients have increased and our workload has also increased, and yet we are still the same number of human resources."* (Site A Clinician)

*"We do have other counselors who do not have proper skills…we can put that as a challenge. Yes, some counselors will not have proper skills maybe on* [contact] *elicitation, they will not manage to…bring out all the their sexual contacts."* (Site B, Healthcare provider)

## Implementation

A perpetual challenge cited was that facility infrastructure and workflow was not conducive to optimizing privacy during HIV and recency result reporting.

*"So, for example, I am in this building, and I am found positive, okay, so recency will not happen there as a result I have to trek to here in my confusion. Should it happen to say we need some lab tests then I will have to trek back to the lab. After results are out again, I have to trek back here…"*. (Site A, Healthcare worker)

In addition, resources for strengthening RTRI result reporting systems at the facility would also need to be factored into future implementation considerations.

*"Just to say that this program has been implemented very well except at a certain point where we had some challenges with documentation. …So, that was corrected and this time we are doing very well."* (Site A Clinician)

## Programmatic considerations for scaling POC RTRI

Resource allocation for POC RTRI will have to balance the utility of testing at the POC as part of national surveillance efforts against the costs of implementation. Programmatic considerations for expanding POC RTRI are summarized in Table 5.

**Table 5. Program considerations for POC RTRI scale up.**

| Area | Consideration | Recommendations |
|---|---|---|
| RTRI assay performance | Relatively high rates of false positive recency results on the RTRI | Utilize RTRI results as part of a RITA |
| Discordant RTRI results | Challenges with test band interpretation | Implement QC measures early into implementation to avoid discrepancies; Implement automated test strip reader to ensure interpretation accuracy |
| Result return and reporting | Will be challenging to perform the RTRI at the POC and not return results or integrate into patient records | Consider social harms that may occur as a result of returning test results to clients; study counselling approaches for safely reporting recency results to clients |
| POC Quality control (QC) | Routine QC, refresher training, proficiency panels, and monitoring of site performance would be needed, particularly in the initial stages of facility-wide rollout | Incorporate RTRI QC into the broader QC for the national HIV testing program |
| Utility | Relatively high rates of false recent results | Consider using RTRI results along with VL (and potentially other tests) as part of a robust RITA, which may decrease the utility of POC RTRI on its own |

Note: RTRI – rapid test for recent infection, RITA – recent infection testing algorithm, POC – point of care.

## Discussion

In this descriptive study, we compared the characteristics of the Asanté RTRI done by health workers at the POC to that done by specialized laboratory workers and found that 96% of the time results at the POC agreed with those from the central laboratory. These quantitative results fit with our qualitative findings suggesting that it is feasible for healthcare workers to conduct the RTRI at the POC in routine HIV care settings in urban Zambia provided the proper training, quality control procedures, and health system resourcing are in place. In so far as the Asanté RTRI can be used to estimate changes in the prevalence of recent infection in specific populations and geographies, it could be leveraged to channel HIV prevention interventions to the people and places at greatest risk for ongoing HIV acquisition and onward transmission [19–21]. Transitioning Asanté RTRI implementation from central laboratories to the POC, where it was designed for use, could help reduce sample courier costs, decongest central laboratories, shorten aggregate result reporting times to public health practitioners, and, theoretically, enable faster identification of potential recent infection "hotspots" for public health action. This would especially be effective if other components of a RITA, such as CD4 counts and VL could similarly be decentralized to the POC.

However, false recent results may severely limit application of the Asanté RTRI for public health purposes or for HIV prevention engaging specific populations. In our study, we observed that 43% of participants with a recent RTRI had a suppressed VL. While reportedly HIV undiagnosed and ART naïve at the time of study enrollment, these individuals may have been aware of their HIV status and already on ART at the time of presenting as "new" to HIV care [22]. Such high rates (15–90%) of false positive recency results have been observed in other countries and pose a challenge for accurately estimating HIV incidence and other cascade indicators [9,23]. Given these challenges, the Asanté RTRI may best be used as part of a RITA, incorporating VL and other relevant clinical and laboratory measures, to enable the accurate identification of truly recent infections for public health programming [20,24,25]. The dependence of the Asanté RTRI on other RITA measures, such as an accompanying VL, may limit the utility of its implementation at the POC.

While it was not the main study aim, we did identify a few instances (about 1% of participants) where the Asanté RTRI results were discrepant with the national HIV diagnostic algorithm. The inclusion of a third diagnostic test in national algorithms, as recently adopted in Zambia, can help improve the quality of HIV testing in the national program. We observed a higher median VL among PLHIV with long-term infection than those with recent infection. More research is needed to determine the VL thresholds and durations that encourage the development of highly avid antibodies for detection by RTRI assays. HIV viremia is an important predictor of HIV transmission risk, particularly early in infection, and, thus, focusing HIV services on individuals with viremia and their partners, irrespective of recency status, can be important for achieving epidemic control [5,26]. As VL testing at the time of HIV care entry scales up in high-burden settings, it will allow better population-level monitoring of VL and identification of individuals with high VLs who contribute disproportionately to transmission. As baseline VL becomes more available in HIV treatment programs, further research will be needed to understand the added value offered by RTRI testing beyond baseline VL testing for recency surveillance and potential "hotspot" identification.

We complemented our quantitative evaluation of POC Asanté RTRI performance by speaking with non-laboratory HCW end users who implemented the RTRI at POC. Facility healthcare workers did not report encountering major burden from adding recency testing to their daily duties, although they did acknowledge an increase in client volumes and paperwork. The majority believed that RTRI testing at the POC was feasible with additional health system resourcing, particularly for human resources to conduct testing. In addition, they believed the Asanté RTRI could be implemented with fidelity and sustained in the health system through continuous on-site training and supportive supervision. This would also support maximal adoption by end users and integration with other HIV testing services. Overall, they felt the addition of POC RTRI was acceptable and appropriate for their practice setting with the support provided by the study and could be acceptable and appropriate under routine operational conditions in Lusaka, Zambia with sustained additional human resources, counselling for clients, and other programmatic inputs and supports.

The Zambian national HIV recent infection surveillance program utilizes data from Asanté RTRI testing done at central laboratories to inform a targeted public health response at sites with a high percentage of RITA recent results in a given district. The operational issues encountered in this study around visual test band interpretation and the high false positive rate for RTRI results compared to the RITA suggest that POC deployment of the Asanté RTRI on its own has limited utility, even when appropriate laboratory QC systems are in place. If used as part of the national RITA, however, POC RTRI testing could have a role if the intrinsic limitations of the RTRI assay—and its implications for the positive predictive value of the assay—are taken into account [27]. Adopting POC Asanté RTRI testing programmatically may prove challenging to withhold RTRI results from individuals, as is done in the national program currently, creating an environment of mistrust about the test specifically, and the recency program more broadly [28–30]. Thus, any consideration of Asanté RTRI testing at the POC should first prompt additional stakeholder consultation, particularly with the communities and populations affected, and careful policy maker consideration to ensure safe result reporting without creating undue social harms.

We acknowledge several study limitations. First, we had a relatively small sample of participants with recent infection, limiting power for statistical comparisons and external validity. However, we report 95% confidence intervals, accounting for these limited observations from a statistical significance perspective, and acknowledge that our results may not hold outside sub-Saharan Africa or urban settings like Lusaka, Zambia. Second, there is increasing evidence that a substantial portion of individuals presenting as new to HIV care have been in HIV care previously [22], which may dilute the target population for recency testing with individuals with chronic HIV infection who have been in care previously. This artifact is reflected in the "real world" analysis undertaken here. However, we have done a sensitivity analysis excluding PLHIV found to have a suppressed baseline viral load that showed improved classification accuracy for the POC RTRI compared to the RITA when these individuals were considered. Finally, we conducted this study at only two pilot sites in urban Lusaka, which both received support from PEPFAR and implementing partners, so these results may not be generalizable to less well-resourced settings in sub-Saharan Africa, such as hard-to-reach rural areas.

In conclusion, the Asanté RTRI can be performed reliably at the POC and is acceptable to non-laboratory healthcare workers operating at government health facilities. Due to non-trivial rates of false recent results, the RTRI should be adopted alongside a VL test as part of a RITA to reduce instances of misclassification. Ultimately, the greatest benefit of the Asanté RTRI will come from its integration into HIV surveillance programs that rapidly leverage quality-assured RITA results for geographically prioritized and differentiated HIV prevention activities.

## Supporting information

**S1 File. Supplementary tables and figures.**
(DOCX)

## Acknowledgments

We are grateful to the study participants, the study teams involved in survey adaptation and collecting and monitoring data collection, and support from the Zambian Ministry or Health without which this work would not be possible.

**Disclaimer:** The findings and conclusions in this report are those of the author(s) and do not necessarily represent the official position of CDC.

## Author contributions

**Conceptualization:** Shilpa S. Iyer, Peter Minchella, Theodora Savory - van Huis, Carolyn Bolton, Michael E. Herce.

**Data curation:** Shilpa S. Iyer, Jake M. Pry, Herbert Kapesa, Misinzo Moono, Chilambwe Mwila, Christiana Frimpong, Mirriam Nanyangwe, Lumbani Phiri, Ruth Ngandu, Precious Sakanya, Sharon Mwansa, Talandila Phiri, Mizinga Haciwa, Patricia Maritim.

**Formal analysis:** Jake M. Pry.

**Funding acquisition:** Carolyn Bolton, Michael E. Herce.

**Investigation:** Shilpa S. Iyer, Jake M. Pry, Chilambwe Mwila, Christiana Frimpong, Mirriam Nanyangwe, Ruth Ngandu, Precious Sakanya, Sharon Mwansa, Talandila Phiri, Mizinga Haciwa, Patricia Maritim, Peter Minchella, Theodora Savory - van Huis, Michael E. Herce.

**Methodology:** Shilpa S. Iyer, Jake M. Pry, Peter Minchella, Theodora Savory - van Huis, Carolyn Bolton, Michael E. Herce.

**Project administration:** Shilpa S. Iyer, Chilambwe Mwila, Christiana Frimpong, Lumbani Phiri, Kemba Lee, Melissa Arons, Tiffiany Aholou, Peter Minchella, Theodora Savory - van Huis, Carolyn Bolton, Michael E. Herce.

**Resources:** Shilpa S. Iyer, Melissa Arons, Michael E. Herce.

**Supervision:** Shilpa S. Iyer, Herbert Kapesa, Misinzo Moono, Chilambwe Mwila, Christiana Frimpong, Lumbani Phiri, Kemba Lee, Michael E. Herce.

**Validation:** Jake M. Pry, Michael E. Herce.

**Visualization:** Jake M. Pry.

**Writing – original draft:** Shilpa S. Iyer, Jake M. Pry, Michael E. Herce.

**Writing – review & editing:** Shilpa S. Iyer, Jake M. Pry, Patricia Maritim, Melissa Arons, Tiffiany Aholou, Peter Minchella, Theodora Savory - van Huis, Carolyn Bolton, Michael E. Herce.

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
