## [Decision Letter · Decision Letter 0]

30 Jan 2026

PONE-D-25-63413The Asanté™ HIV-1 Rapid Recency® Assay is reliable, feasible, and acceptable for use at the point-of-care in Lusaka, Zambia.PLOS One

Dear Dr. Pry,

Thank you for submitting your manuscript to PLOS ONE. After careful consideration, we feel that it has merit but does not fully meet PLOS ONE’s publication criteria as it currently stands. Therefore, we invite you to submit a revised version of the manuscript that addresses the points raised during the review process.

**ACADEMIC EDITOR:**The manuscript has strong potential and addresses an important HIV surveillance issue in sub-Saharan Africa. However, key clarifications, improved methodological transparency, and a more cautious, consistent interpretation of findings are needed before acceptance is possible.The manuscript has strong potential and addresses an important HIV surveillance issue in sub-Saharan Africa. However, key clarifications, improved methodological transparency, and a more cautious, consistent interpretation of findings are needed before acceptance is possible.

**Essential Revisions**

Clarify RITA use as reference standard and discuss potential incorporation bias.Add more detail on VL testing selection and potential biases.Strengthen qualitative methodology reporting (coding process, saturation, inter‑coder reliability).Expand the discussion on generalizability limitations.Provide more systematic explanation of false recency results.Tighten Introduction and Discussion to improve conciseness.Expand reporting of QC procedures and implications for real-world implementation.Improve clarity of figures/tables and ensure consistent use of terms.

**Desirable Revisions**

Add quotes to support qualitative themes more robustly.Merge repetitive statistical explanations.Add confidence intervals to all key metrics.Consider including cost/resource implications in greater detail.

If applicable, we recommend that you deposit your laboratory protocols in protocols.io to enhance the reproducibility of your results. Protocols.io assigns your protocol its own identifier (DOI) so that it can be cited independently in the future. For instructions see: https://journals.plos.org/plosone/s/submission-guidelines#loc-laboratory-protocols. Additionally, PLOS ONE offers an option for publishing peer-reviewed Lab Protocol articles, which describe protocols hosted on protocols.io. Read more information on sharing protocols at . Additionally, PLOS ONE offers an option for publishing peer-reviewed Lab Protocol articles, which describe protocols hosted on protocols.io. Read more information on sharing protocols at https://plos.org/protocols?utm_medium=editorial-email&utm_source=authorletters&utm_campaign=protocols..

We look forward to receiving your revised manuscript.

Kind regards,

Hamufare Dumisani Mugauri, Ph.D. Epidemiology and Public Health

Academic Editor

PLOS One

5. We notice that your supplementary figure and table are included in the manuscript file. Please remove them and upload them with the file type 'Supporting Information'. Please ensure that each Supporting Information file has a legend listed in the manuscript after the references list.

Reviewers' comments:

Reviewer's Responses to Questions

**Comments to the Author**

1. Is the manuscript technically sound, and do the data support the conclusions?

Reviewer #1: Yes

Reviewer #2: Yes

2. Has the statistical analysis been performed appropriately and rigorously? 

Reviewer #1: Yes

Reviewer #2: Yes

3. Have the authors made all data underlying the findings in their manuscript fully available?

Reviewer #1: No

Reviewer #2: Yes

4. Is the manuscript presented in an intelligible fashion and written in standard English?

Reviewer #1: Yes

Reviewer #2: Yes

5. Review Comments to the Author

Reviewer #1: This is a valuable and timely study that addresses an important gap in HIV recency testing at point-of-care in a resource-limited setting. The manuscript presents a robust mixed-methods evaluation of the Asant HIV-1 Rapid Recency Assay (RTRI) at point-of-care (POC) in Lusaka, Zambia, demonstrating strong concordance with Central Laboratory testing and feasibility among healthcare workers. Minor revisions are needed to enhance clarity, generalizability, and methodological precision, thereby improving readiness for publication.

Comments:

General: The author(s) declare in the financial disclosure section that no specific funding was received for this work. However, in lines 130, 495, and 521, the author (s) acknowledge the financial support from PEPFAR through the CDC.

Abstract:

In the methods section, the phrase “returned to both sites and clients” (line 43) is ambiguous and could be misinterpreted as a standard practice or a primary outcome of the study. Based on the manuscript context, it seems results were made available to clients at follow-up visits as part of the study protocol. Consider revising the sentence to reflect the availability of results to clients at scheduled study follow-up visits as part of a supervised counselling process.

The phrase “health staff” in the sentence “four focus group discussions (FGDs) with health staff explored perceptions of POC-RTRI implementation” (lines 45-46) is broad and could include anyone from administrators to laboratory technicians. Please consider specifying the cadres who participated in the FDGs. For consistency and transparency, please ensure to explicitly state the cadres in the Methods section under “Qualitative Data Collection and Analysis”.

Discrepancy in the details in the abstract and results sections: In the abstract (lines 49-50, it states: “The POC-RTRI identified fewer recent infections than laboratory RTRO (85% vs 100.0%). However, in the main results (lines 264-275), the concordance table shows 27 recent at POC vs 35 recent at the Central laboratory, which is 77.1%, not 85%. Please clarify this discrepancy and ensure consistency between the abstract and the results sections. Additionally, the abstract reports 96.0 agreement between POC and laboratory RTRI, while Table 1 shows an agreement of 96.5. These likely reflect rounding/precision differences but require harmonization. Please consider providing an accurate agreement.

The Kappa value in the abstract (0.821) matches the main text Kappa for N=317 (0.812 on page 15 is likely a typo; should it be 0.821 as in the abstract?). Please clarify.

Methods:

Sample size: The methods section in the abstract and main text do not state the study’s sample size, which is a standard and necessary element for readers to assess the scale of the work. The sample size is first explicitly mentioned in the results section, on line 243 of the manuscript. Please consider adding the total number of enrolled participants for completeness.

Laboratory and test procedures: While the manuscript notes that telemonitoring was used to clarify faint band lines (line 190), the specific decision rule or visual threshold for interpreting faint Long Term lines is not described. To ensure methodological reproducibility and clarify the source of interpretation errors, please explicitly state the criteria used to determine whether a faint LT was recorded as present (long-term) or absent (recent). For example, was any visible line considered positive, or was a minimum threshold applied?

Data availability statement: Please consider depositing your data in a repository to obtain a citable DOI for the dataset linked to your manuscript. You can indicate the DOI in the statement, “De-identified and aggregated study data will be made publicly available through Data Dryad (doi: [pending]).

Results: Please consider expanding Table 3 to include the positive and negative predictive values that would help answer the question, “If this test result shows that the person was infected recently, how likely is that to be true?

Discussion: Consider linking the high false recent rate to the sensitivity analysis, which shows near-perfect specificity when excluding VL.

Reviewer #2: - Introduction:

i. Acknowledge the work of other researchers on the topic and identify the gaps in knowledge your study will fill

-Objectives:

i. Clearly describe General and specific objectives of your study

- Methods section needs improvement

i. For quantitative component 321 participants were recruited, how this sample size was calculated and what was sampling method. Non random sampling might have introduced bias in the study

ii. Potential participants were approached by study staff for participation. What was refusal rate ?

iii. Written informed consent was obtained from the participants. Why written consent ? that might have prevented members of key populations to participate, that might have introduced bias in the study

iv. Locator form, enrolment demographic case reporting form and bio-behviroal information of study participats was documented. Please describe data collection instrument.

6. PLOS authors have the option to publish the peer review history of their article (what does this mean?). If published, this will include your full peer review and any attached files.). If published, this will include your full peer review and any attached files.

.

Reviewer #1: No

Reviewer #2: **Yes:**Dr. Sharaf Ali ShahDr. Sharaf Ali Shah

---

## [Author Response · Author response to Decision Letter 1]

18 Mar 2026

Dear Dr. Mugauri,

Many thanks for reviewing our revised manuscript (PONE-D-25-63413) entitled: “The Asanté™ HIV-1 Rapid Recency® Assay is reliable, feasible, and acceptable for use at the point-of-care in Lusaka, Zambia.” We also appreciate the additional time to make edits in accordance with the insightful feedback.

On behalf of my colleagues, I wanted to thank you and the reviewers for the thoughtful review and the additional time to make edits accordingly. We have considered each comment carefully and have made the changes noted below (in blue font), which we believe have strengthened the manuscript. We have also undertaken additional revisions to improve the overall flow and readability of the manuscript. We have enclosed two versions of the revised manuscript in Word (.docx)—one with changes highlighted under “track changes” and a second “clean” version with all changes accepted.

Please find here our point-by-point responses to reviewer comments, organized by headings taken from the reviewers provided on 30 January 2026 (also, please note that for ease of reference, the lines below refer to the “clean” version of the revised manuscript):

Begin itemized review response.

Editor’s comments:

The manuscript has strong potential and addresses an important HIV surveillance issue in sub-Saharan Africa. However, key clarifications, improved methodological transparency, and a more cautious, consistent interpretation of findings are needed before acceptance is possible.

Essential Revisions

Clarify RITA use as reference standard and discuss potential incorporation bias.

Response: Thank you for this note, it is true that some incorporation bias is involved here and have added language to that end in the limitations section.

“It is possible that incorporation bias played a role in our analysis, given that the recent infection testing algorithm requires the rapid test for recent infection. However, this bias tends to strengthen the relationship against the standard and bias the comparison toward the null.”

Add more detail on VL testing selection and potential biases.

Response: Thank you for this note, we have added information in the methods about the selection process and discussed potential bias in the limitations section.

“Potential participants included all individuals accessing routine HTS and newly diagnosed with HIV at a study clinic. These potential participants were invited by study staff to take part in the study and provide a blood sample for viral load testing after consent.”

Strengthen qualitative methodology reporting (coding process, saturation, inter‑coder reliability).

Response: We have added a statement outlining that we applied the six steps to qualitative analysis as proposed by Braun and Clarke. Given that we were working more inductively and shaped by the data itself, we reviewed all the transcripts from the discussions conducted. Therefore, saturation was not a key consideration. Coder reliability approaches where intercoder reliability scores are required represent one of many approaches to thematic analysis but it was not the one that we applied in our work. We used a more reflexive thematic analysis approach which is more iterative and interpretative to account for the data from the health facilities and our codebook that was generated from Proctor outcome’s framework. As Braun and Clarke outline, such approaches are useful in applied research where “a codebook is used not for the purposes of determining the reliability and accuracy of coding but to chart or map the developing analysis.”

Expand the discussion on generalizability limitations.

Response: Thank you, we have amended language accordingly.

“First, we had a relatively small sample of participants with recent infection, limiting power for statistical comparisons and external validity. However, we report 95% confidence intervals to present the uncertainty surrounding the limited observations from a significance perspective and acknowledge that our results may not hold outside urban settings in sub-Saharan Africa, urban settings like Lusaka, Zambia.”

Provide more systematic explanation of false recency results.

Response: Thank you, we have expanded accounting of false recency results in the methods section as follows:

“A recent result on the RTRI accompanied by a suppressed viral load was considered as a false-recent result and counted as long-term.”

Tighten Introduction and Discussion to improve conciseness.

Response:

This comment is noted and appreciated. However, given the subtle nature of our findings, and the limitations in sample size we believe that the additional language helps bring out the nuance in the study set-up and discussion.

Expand reporting of QC procedures and implications for real-world implementation.

Response: Thank you for this note, we have expanded the description of QC reporting in the methods“ lines 180-184 as follows: “The lines drawn on the picture were verified to ensure they corresponded to the result documented through regular quality control (QC) record reviews done by the research assistants and study laboratory specialist during supportive supervision visits.”

Improve clarity of figures/tables and ensure consistent use of terms.

Response: Thank you, we have uploaded full resolution versions of the figures in a separate file to avoid picture file compression in the Word document.

Desirable Revisions

Add quotes to support qualitative themes more robustly.

Response: We have provided additional quotations to support the narratives and illuminate qualitative themes. Additionally, we have expanded upon selected quotes to more fully reflect upon these themes in the text.

Merge repetitive statistical explanations.

Response: Thank you for this guidance, we have revised the results to reduce redundancies.

Add confidence intervals to all key metrics.

Response: Thank you for this note, confidence intervals added where appropriate.

Consider including cost/resource implications in greater detail.

Response: Thank you for this suggestion, perhaps this will be considered for future work – it is outside the scope of this manuscript.

Reviewer #1:

This is a valuable and timely study that addresses an important gap in HIV recency testing at point-of-care in a resource-limited setting. The manuscript presents a robust mixed-methods evaluation of the Asant HIV-1 Rapid Recency Assay (RTRI) at point-of-care (POC) in Lusaka, Zambia, demonstrating strong concordance with Central Laboratory testing and feasibility among healthcare workers. Minor revisions are needed to enhance clarity, generalizability, and methodological precision, thereby improving readiness for publication.

Response: Many thanks for taking the time to review and provide thoughtful feedback, we have done our best to revise in response to your insights.

Comments:

General: The author(s) declare in the financial disclosure section that no specific funding was received for this work. However, in lines 130, 495, and 521, the author (s) acknowledge the financial support from PEPFAR through the CDC.

Response: Thank you for this note, we have updated the financial disclosure section accordingly.

Abstract:

In the methods section, the phrase “returned to both sites and clients” (line 43) is ambiguous and could be misinterpreted as a standard practice or a primary outcome of the study. Based on the manuscript context, it seems results were made available to clients at follow-up visits as part of the study protocol. Consider revising the sentence to reflect the availability of results to clients at scheduled study follow-up visits as part of a supervised counselling process.

Response: Thank you for noting this – we have revised to improve clarity as follows.

“Final recent infection testing algorithm (RITA) results were returned to the study sites and were made available to clients upon follow-up visit”

The phrase “health staff” in the sentence “four focus group discussions (FGDs) with health staff explored perceptions of POC-RTRI implementation” (lines 45-46) is broad and could include anyone from administrators to laboratory technicians. Please consider specifying the cadres who participated in the FDGs. For consistency and transparency, please ensure to explicitly state the cadres in the Methods section under “Qualitative Data Collection and Analysis”.

Response: Thank you, we have revised the qualitative section of the methods to introduce the cadre classification.

“The HCW cadres were classified as peer treatment supporters (also known as community health workers), nurses, clinical officers, HIV testing service counselors, and laboratory technicians.”

Discrepancy in the details in the abstract and results sections: In the abstract (lines 49-50, it states: “The POC-RTRI identified fewer recent infections than laboratory RTRO (85% vs 100.0%). However, in the main results (lines 264-275), the concordance table shows 27 recent at POC vs 35 recent at the Central laboratory, which is 77.1%, not 85%. Please clarify this discrepancy and ensure consistency between the abstract and the results sections. Additionally, the abstract reports 96.0 agreement between POC and laboratory RTRI, while Table 1 shows an agreement of 96.5. These likely reflect rounding/precision differences but require harmonization. Please consider providing an accurate agreement.

Response: thank you for this astute note, we have revised accordingly.

“The POC-RTRI results indicated more recent infections than laboratory-RTRI (30 vs 27) though the discrepancy was POC-RTRI false positive (85.0% compared to 100.0% sensitivity against RITA)”

And

“Agreement between POC and laboratory RTRI was 96.5%, with a Kappa of 0.821 (95% CI: 0.713–0.928).”

The Kappa value in the abstract (0.821) matches the main text Kappa for N=317 (0.812 on page 15 is likely a typo; should it be 0.821 as in the abstract?). Please clarify.

Response: Thank you for this note, the discrepancy has been resolved as 0.812.

“Agreement between POC and laboratory RTRI was 96.5%, with a Kappa of 0.812 (95% CI: 0.704–0.920).”

Methods:

Sample size: The methods section in the abstract and main text do not state the study’s sample size, which is a standard and necessary element for readers to assess the scale of the work. The sample size is first explicitly mentioned in the results section, on line 243 of the manuscript. Please consider adding the total number of enrolled participants for completeness.

Response: We did not specify a sample size for this descriptive, pilot implementation research study that did not involve a priori hypothesis testing. Rather, we estimated a feasible recruitment target based on CIDRZ routine program data for the duration of the planned study. As such, we estimated that we would identify a total of 50 new HIV-positive clients per week across both sites, of whom approximately 50% would meet study eligibility criteria, leaving an estimated recruitment target of 600 over a 24 week pilot study period. Due to unforeseen challenges that emerged because of the COVID-19 pandemic, clinic visit volumes reduced by roughly 50% during the study period, leaving us with the current study enrollment of about 300 participants.

Laboratory and test procedures: While the manuscript notes that telemonitoring was used to clarify faint band lines (line 190), the specific decision rule or visual threshold for interpreting faint Long Term lines is not described. To ensure methodological reproducibility and clarify the source of interpretation errors, please explicitly state the criteria used to determine whether a faint LT was recorded as present (long-term) or absent (recent). For example, was any visible line considered positive, or was a minimum threshold applied?

Response: Thank you for your comment and suggestion. We have included a description of the adjudication process for clarity, adding that any observed band, even a faint one, was recorded as present and the sample was considered as long term for this study.

Data availability statement: Please consider depositing your data in a repository to obtain a citable DOI for the dataset linked to your manuscript. You can indicate the DOI in the statement, “De-identified and aggregated study data will be made publicly available through Data Dryad (doi: [pending]).

Response: Thank you for this note, we have added the DOI reference.

“Data Availability

De-identified and aggregated study data will be made publicly available through Data Dryad upon publication (doi: 10.5061/dryad.k98sf7mnf).”

Results: Please consider expanding Table 3 to include the positive and negative predictive values that would help answer the question, “If this test result shows that the person was infected recently, how likely is that to be true?

Response: Thank you for this note. We have opted to limit this table 3 analysis to test characteristics rather than population-dependent characteristics like predictive values.

Discussion: Consider linking the high false recent rate to the sensitivity analysis, which shows near-perfect specificity when excluding VL.

Response: We believe the current description of the sensitivity analysis is sufficient to allow readers to make their own determinations about the classification accuracy of POC RTRI and laboratory-based RTRI.

Reviewer #2

Introduction

Acknowledge the work of other researchers on the topic and identify the gaps in knowledge your study will fill

Response: Thank you for this comment we have revised accordingly.

“In theory, adoption of RTRI at the POC could enable faster time to results and public health action at facility-level, while saving on resources for sample shipping and specialized laboratory personnel. However, no published studies from sub-Saharan Africa (SSA) have examined the effects of the Asanté RTRI when used at the POC by non-laboratory healthcare workers on testing accuracy, reliability, or site-level operations.”

Objectives

Clearly describe General and specific objectives of your study

Response: Thank you for this note. We have included this information.

“In this study, we sought to assess the operational feasibility and reliability of the Asanté RTRI when used by non-laboratory healthcare workers at the POC, and to evaluate the frequency of discrepant results by RTRI testing setting (i.e., POC versus centralized laboratory). Using additional laboratory results, including VL testing, we also evaluated the field performance of the Asanté RTRI when used in the central laboratory and at the POC compared to the program reference standard used in Zambia, the central laboratory-based RITA. Lastly, through focus group discussions (FGDs) with frontline healthcare workers (HCW), we sought to assess the feasibility of implementing Asanté RTRI at the POC in Zambian health facilities.”

Methods section needs improvement

For quantitative component 321 participants were recruited, how this sample size was calculated and what was sampling method. Non random sampling might have introduced bias in the study

Response: Thank you for this note, we have included sample size calculation information in methods. We did not specify a sample size for this descriptive, pilot implementation research study that did not involve a priori hypothesis testing. Rather, we estimated a feasible recruitment target based on CIDRZ routine program data for the duration of the planned study. As such, we estimated that we would identify a total of 50 new HIV-positive clients per week across both sites, of whom approximately 50% would meet study eligibility criteria, leaving an estimated recruitment target of 600 over a 24 week pilot study period. Due to unforeseen challenges that emerged because of the COVID-19 pandemic, clinic visit volumes reduced by roughly 50% during the study period, leaving us with the current study enrollment of about 300 participants.

Potential participants were approached by study staff for participation. What was refusal rate ?

Response: We report these numbers in an accompanying main effects paper currently under review at the Journal of the International AIDS Society and have added brief mention

---

## [Editor Report · Decision Letter 1]

25 Mar 2026

The Asanté™ HIV-1 Rapid Recency® Assay is reliable, feasible, and acceptable for use at the point-of-care in Lusaka, Zambia.

PONE-D-25-63413R1

Dear Dr. Pry,

We’re pleased to inform you that your manuscript has been judged scientifically suitable for publication and will be formally accepted for publication once it meets all outstanding technical requirements.

An invoice will be generated when your article is formally accepted. Please note, if your institution has a publishing partnership with PLOS and your article meets the relevant criteria, all or part of your publication costs will be covered. Please make sure your user information is up-to-date by logging into Editorial Manager at Editorial Manager® and clicking the ‘Update My Information' link at the top of the page. For questions related to billing, please contact  and clicking the ‘Update My Information' link at the top of the page. For questions related to billing, please contact billing support..

Kind regards,

Hamufare Dumisani Mugauri, Ph.D. Epidemiology and Public Health

Academic Editor

PLOS One
---

## [Editor Report · Acceptance letter]

PONE-D-25-63413R1

PLOS One

Dear Dr. Pry,

I'm pleased to inform you that your manuscript has been deemed suitable for publication in PLOS One. Congratulations! Your manuscript is now being handed over to our production team.

Kind regards,

on behalf of

Dr Hamufare Dumisani Mugauri

Academic Editor

PLOS One